# Characterization and In Vitro Fungicide Sensitivity of Two *Fusarium* spp. Associated with Stem Rot of Dragon Fruit in Guizhou, China

**DOI:** 10.3390/jof9121178

**Published:** 2023-12-08

**Authors:** Jin Zhao, Miao Huang

**Affiliations:** 1College of Agriculture, Guizhou University, Guiyang 550025, China; 2059hm@sina.com; 2Institute of Pepper Industry and Technology, Guizhou University, Guiyang 550025, China

**Keywords:** dragon fruit, stem rot, *Fusarium*, identification, fungicides

## Abstract

Dragon fruit (*Hylocereus polyrhizus*) constitutes an important economic industry in Guizhou Province, China, and the occurrence of stem rot has become increasingly severe. In this study, we aimed to determine the causative pathogens of stem rot in this region and analyze their sensitivity to fungicides. Twenty-four fungal isolates were obtained from diseased tissues, from which H-4 and H-5 were confirmed as pathogens based on Koch’s postulates. Based on the morphological characteristics of macroconidia, microconidia, and colony morphology, the polygenic phylogenetic tree constructed using internal transcribed spacer, elongation factor 1-alpha, and retinol-binding protein-2 gene fragments, and carbon source metabolism analysis using FF microplates, the two pathogens were identified as *F. oxysporum* and a newly discovered pathogen, *F. concentricum*. In addition, the in vitro toxicity of eight fungicides against both pathogens was measured based on the mycelial growth rate. The results showed that trifloxystrobin 25%·tebuconazole 50% (75 WG) exhibited the strongest inhibitory effect against both isolates, with EC50 values of 0.13 µg/mL and 0.14 µg/mL, respectively. These findings hold significant potential for guiding the effective treatment of stem rot in dragon fruit in Guizhou, China.

## 1. Introduction

Dragon fruit or pitaya (*Hylocereus polyrhizus*) is a tropical and subtropical herbaceous fruit tree that belongs to the family Cactaceae and the genus *Hylocereus* [1]. The fruit enjoys popularity among consumers worldwide and holds substantial economic value because it is rich in iron, phosphorus, and other trace elements, as well as pulp fiber and carotene [2,3]. Originating in Central America, the cultivation of dragon fruit has gradually expanded since the 1990s to numerous provinces in southern China, including Guizhou Province, where small-scale cultivation began in 2001 [4]. The dragon tree swiftly gained prominence as an essential asset for local development owing to its suitability for open-air planting, resistance to barren conditions and drought, and high economic value. In 2020, the cultivated area in Guizhou Province exceeded 6700 ha, and its output reached 80,000 tons, ranking among the highest in China [5]. However, the expansion of cultivation areas, coupled with rising temperatures and increased rainfall, has led to a notable surge in the occurrence of fungal diseases affecting dragon fruit in Guizhou, ultimately affecting their economic benefits [6,7].

Dragon fruit is susceptible to over seven fungus-related diseases, including canker and anthracnose, with stem rot being prevalent [8,9,10]. Stem rot has become widespread across all planting regions in Guizhou Province, resulting in significant yield reduction and, in severe cases, crop failure [11]. This disease is easily disseminated through wind or rain, specifically during hot and rainy weather conditions [12]. The symptoms of this disease progress in distinct stages, starting with small brown patches on the stem that gradually spread throughout the plant. As the disease advances, the late-stage lesion transitions from green to a dark yellow color, developing penetrating and translucent characteristics with soft tissue rot [13]. Currently, there are conflicting reports regarding the pathogens responsible for stem rot in China. Several fungi, including *Enterbacter* sp., *Bipolaris cactivora*, *Neoscytalidium dimidiatum*, *Fusarium semitectum*, *F. oxysporum*, and *F. moniliforme* are speculated to be the major contributors [8,14,15,16]. Establishing pathogen identification is essential for effective disease management for both farmers and plant caretakers. Therefore, a comprehensive understanding of the specific pathogenic species associated with local stem rot is critical.

Despite the adverse environmental effects associated with the use of fungicides, they remain the most effective method for controlling many plant diseases [17,18]. Several active molecules, such as triazoles (FRAC 3) and strobilurins (FRAC 11), with high efficiency, safety, and broad-spectrum fungal inhibition have been reported as the most effective fungicides currently available against a wide range of pathogens, such as basidiomycetes, oomycetes, and ascomycetes, including *Fusarium* species [19,20,21]. The triazole family, including tebuconazole and difenoconazole, belonging to the demethylase inhibitor group of fungicides, have the ability to prevent pathogens from disrupting ergosterol biosynthesis [22,23]. Strobilurin fungicides, including azoxystrobin and trifloxystrobin, can prevent pathogens from inhibiting mitochondrial respiration [24].

However, many studies have reported an increase in the fungicide-resistant strains of several types of pathogens [25]. Notably, the major plant pathogen *F. oxysporum*, which affects tomatoes, potatoes, dragon fruit, and other commercial crops, has demonstrated resistance to various fungicides, including difenoconazole and tebuconazole [26,27]. Understanding the sensitivity of pathogens to various fungicides and employing appropriate doses are critical for effective chemical control of the disease, as well as environmental protection. However, the pathogens responsible for stem rot in Guizhou, China, and their sensitivity to fungicides remain unclear.

In this study, we obtained stem rot samples from dragon fruit in Guizhou Province, isolated the fungi from affected tissues, and used Koch’s postulates to confirm their etiology. To ensure accurate identification of the pathogens, we integrated the traditional morphological characterization approach and the modern molecular multigene phylogenetic tree method. Subsequently, the sensitivity of several fungicides was determined in the laboratory, and the optimum inhibitor was identified. These findings can aid in the prevention and control of pitaya stem rot not only in the Guizhou Province but also in the neighboring regions.

## 2. Materials and Methods

### 2.1. Collection, Isolation, and Conservation

In total, eight stem rot branch samples were collected from dragon fruit cultivation areas in Zhenfeng and Luodian, Guizhou Province. The margins of diseased and healthy tissues were cut into 5 mm × 5 mm pieces, followed by disinfection with 0.1% mercury for 30 s. Subsequently, they were immersed in 75% ethanol for 5 s, rinsed thrice with sterile water, and dried on a sterile absorbent paper. The isolates were obtained by incubating the cleaned disease samples at 26 °C for 2–3 d at the center of a potato glucose agar (PDA) medium plate 9 cm in diameter. Next, they were purified using marginal mycelia or single hyphal culture methods. All isolates were stored in freezer tubes containing 20% (*v*/*v*) glycerol at −20 °C in the dark.

### 2.2. Koch’s Postulates

The pure fungal strains were inoculated on a PDA or oatmeal agar (OA) medium and cultivated at 26 °C for 7 d to generate a substantial quantity of conidia. The conidia were adequately collected by rinsing the colony multiple times with sterile water. The conidial concentration was measured using a hemocytometer and subsequently adjusted to 1 × 10^6^ conidia/mL with sterile water. Twelve stem blocks were cut from a healthy dragon fruit and transplanted into 1.8 L pots filled with deionized water-saturated sterilized substrate soil. The spore suspension (200 µL) was then shallowly injected into the healthy dragon fruit branches and moistened with absorbable cotton after approximately a week of stem growth. The control treatments were inoculated with equal amounts of sterile water. Each treatment was repeated three times. Finally, all treatment samples were planted in an artificial climate chamber with a 12/12 h light/dark cycle, a relative humidity of approximately 80%, and a temperature of 26 °C. After 21 d, disease incidence was observed and recorded, and the diseased roots were placed on the PDA medium for pathogens to be re-isolated and identified.

### 2.3. Morphological Studies

Microscopic characteristics such as colony shape and color were visually observed on a 7 d old PDA culture. Moreover, microscopic characteristics such as conidia shape and size were observed using a microscope after culturing on a slide. A 1 cm^3^ cube of OA medium containing mycelia was cut and placed on a sterile slide, and another clean 1 cm^3^ OA medium cube was placed right next to it, with a cover glass placed on top of them. The medium cubes were then placed in a Petri dish covered with sterile filter paper and moistened with a small amount of sterile water. Finally, the Petri dishes were sealed and cultured at 26 °C for approximately 7 d. The conidial characteristics of the slides and cover slides were observed under a microscope. The conidia were photographed using a scanning electron microscope.

### 2.4. DNA Extraction, PCR Analysis, and Multi-Locus Phylogeny

Genomic DNA was extracted from the mycelia of all isolates using a Rapid Fungal Genomic DNA Isolation Kit (B518229; Sangon Biotech, Shanghai, China) according to the manufacturer’s instructions. Sequences of the ribosomal DNA internal transcribed spacer (ITS) region (ITS4: 5′-GGAAGTAAAAGTCG-TAACAAGG-3′, ITS5: 5′-TCCTCCGCTTATTGATATGC-3′), translation elongation factor 1-alpha–encoding gene (*EF-1α*) (EF1: 5′-ATGGGTAAGGARGACAAGAC-3′, *EF2*: 5′-GGARGTACCAGTSATCATGTT-3′), and the second largest subunit of the RNA polymerase II–encoding gene (retinol-binding protein 2 [*RPB2*]) (rpb2-5f: 5′-GAYGAYMGWGATCAYTTYGG-3′, rpb2-7cr: 5′-CCCATRGCTTGYTTRCCCAT-3′) were used for species-level identification [28,29,30].

A Rapid Fungi Genomic DNA Isolation kit (B518229, Sangon Biotech, Shanghai, China) was used to prepare the PCR mixes, which had a total volume of 25 μL, containing 1× Phanta Max Buffer, 1 U DNA Polymerase, 0.2 mM dNTPs, and 0.2 μM of each primer. Next, amplification was conducted using a PCR instrument (846-x-070-723, Analytik Jena, Gottingen, Germany) with the following conditions: denaturation at 94 °C for 3 min and 35 cycles of denaturation at 94 °C for 15 s; annealing at 55 °C (ITS), 60 °C (*EF-1α*), or 51 °C (*RPB2*) for 15 s; and extension at 72 °C for 1 min, followed by a final extension at 72 °C for 5 min. A 2% agarose gel containing a nucleic acid dye was used to segregate the amplification products, which were subsequently photographed under UV light. The validated PCR products were sent to Sangon Biotech for Sanger sequencing. The original sequence was downloaded, analyzed, integrated, and submitted to GenBank. Phylogenetic trees were constructed via the CIPRES web portal using the maximum likelihood (ML) method employing the combined ITS, *EF-1α*, and *RPB2* dataset (https://www.phylo.org/portal2/login!input.action, accessed on 12 September 2023) (Appendix A). The species *F. sacchari* was selected as the outgroup. The RAxML-HPC BlackBox v.7.2.8 was used with its default parameters for the ML analysis.

### 2.5. Metabolic Profiling in the Biolog FF MicroPlate

For species identification, metabolic analysis of fungi was performed on Biolog FF microplates (catalog #1006, Biolog 21124; Cabot Boulevard, Hayward, CA, USA) containing 95 different carbon sources [31]. The isolates were inoculated on a PDA medium at 26 °C for 7 d to generate a considerable quantity of mycelia. The mycelia were gently dipped in a sterile cotton swab and transferred to a test tube containing FF-IF solution. The absorbance of fungal suspension was adjusted to 75 ± 2% at 490 nm using a turbidimeter. A 100 μL suspension was then added to each well of the Biolog FF microplate and incubated at 26 °C. The microplates were removed every 12 h, placed in a MicroStation readout, and measured at 490 nm (mitochondrial activity) and 750 mm (mycelium growth) using the MicrologTM 3 software.

### 2.6. Fungicide Sensitivity Testing

Eight commercial fungicides commonly used in dragon fruit orchards were selected for the in vitro toxicity analysis of strains H-4 and H-5, including azoxystrobin 250 g/L SC, difenoconazole 125 g/L·azoxystrobin 200 g/L (325 g/L SC), difenoconazole 150 g/L·propiconazole 150 g/L (300 g/L SC), and fludioxonil 25 g/L SC (Syngenta Nantong Crop Protection Co., Ltd., Nantong, China); difenoconazole 40% SC (Shandong Dongtai Agrochemical Co., Ltd., Liaocheng, China); trifloxystrobin 25%·tebuconazole 50% (75 WG) (Bayer Aktiengesellschaft, Chengdu, China); carbendazim 50% WP (Sichuan Guoguang Agrochemical Co., Ltd., Chengdu, China); and lime sulfur 45% WP (Hebei Shuangji Chemical Co., Ltd., Xinji, China). Five concentrations of the PDA medium were established by dissolving and diluting each fungicide with sterile water and then absorbing and adding various volumes of the liquid to the PDA medium. All prepared solutions were poured into sterile Petri dishes that were 9 cm in diameter. The center of each dish was inoculated with a 5 mm diameter mycelium disk cut using a hole punch. A PDA medium without fungicides was used as the control. Each treatment was replicated thrice. All dishes were cultured for 8 d at 26 °C, and the colony diameter was measured and recorded using the cross method.

The inhibitory effect of each fungicide on the growth of each isolate was calculated as follows:Inhibitory rate (%) = [(Φc − Φt)/(Φc − 5)] × 100%(1)
where Φc is the diameter of the treated pathogen colony, Φt is the diameter of the pathogen colony in the control group, and 5 is the diameter of the inoculated mycelial disk. The inhibitory rates of the tested fungicides were calculated by Microsoft Excel 2019. The concentration that inhibits fungal growth at 50% (EC_50_), regression equation, and correlation coefficient (r) were analyzed by GraphPad Prism 7 software.

## 3. Results

### 3.1. Pathogenicity Test of the Isolated Strains

Twenty-four fungal isolates were obtained from eight dragon fruit stem rot samples. Based on their morphological characteristics and pretest for pathogenicity, two isolates, H-4 and H-5, were further screened for pathogenicity using Koch’s postulates. After 21 d of artificial inoculation with H-4 and H-5 conidia, noticeable brown rot occurred, which was consistent with the field symptoms (Figure 1). However, plants in the control group exhibited no disease symptoms (Figure 1). The pathogens were isolated and confirmed to be consistent in the diseased tissue of all inoculated plants. Therefore, isolates H-4 and H-5 were confirmed as the pathogens responsible for stem rot in dragon fruit.

### 3.2. Isolate Identification

The pathogenic strains H-4 and H-5 were cultured on a PDA medium and slides of OA medium for morphological identification. These two isolates exhibited rapid growth on PDA, with branched mycelium containing septa. The aerial mycelium appeared white, with a creamy to yellowish pigment (Figure 2A and Figure 3A). Conidial characteristics were observed on the slides. For the H-4 strain, the microconidia were oval with either no septa or one septum, measuring 20.14-(11.65)-8.53 × 6.16-(4.65)-2.54 μm (Figure 2B,C). Macroconidium typically possessed three septa, which were slender, sickle-shaped to nearly straight, with a diameter of 35.34-(23.32)-18.16 × 4.89-(3.87)-3.10 μm (Figure 2D–F). For the H-5 strain, the microconidia were oval with either no septa or one septum, measuring 14.34-(9.65)-7.53 × 6.06-(4.15)-2.04 μm (Figure 3B,C). Macroconidia were observed with 3 to 5 septa, with a diameter of 65.43-(43.53)-36.76 × 4.83-(3.67)-3.03 μm (Figure 3D,E).

For molecular identification, three distinct fragments of ITS, EF-1α, and RPB2 from strains H-4 and H-5 were sequenced and subsequently submitted to the NCBI database (Appendix A). The corresponding sequences of related species were collected, including that of *F. sacchari* as an outgroup. A phylogenetic tree was constructed using the ML method. The results indicated that the H-4 isolate exhibited the least connection to the phylogenetic cluster containing *F. oxysporum*, with a confidence rate exceeding 97% (Figure 4A). The H-5 isolate was located on the same branch as *F. concentricum* (Figure 4B) and demonstrated a 100% support rate.

The metabolic abilities of both isolates were assessed using a Biolog FF MicroPlate, which encompasses 95 different carbon sources. The H-4 isolate demonstrated proficiency in utilizing 74 carbon sources for full growth. However, it demonstrated incapacity in utilizing 21 carbon sources, including dextrin, gentiobiose, A-cyclodextrin, and ß-cyclodextrin, resulting in no growth. The H-5 isolate was responsive to 85 carbon sources but showed insensitivity to 10, including gluconamide, adenosine-5’monophosphate, ltose, and maltitol (Table 1). According to the FF microplate identification result, the two pathogens were *Fusarium* species, and the SIM values were 0.41 and 0.59 for H-4 and H-5, respectively.

Therefore, based on the analysis of morphology, molecular systematics, and metabolic characteristics, H-4 and H-5 were identified as *F. oxysporum* and *F. concentricum*, respectively. Notably, this study is the first to identify *F. concentricum* as a causative pathogen of stem rot in dragon fruit.

### 3.3. Fungicidal Efficacy on the Isolates

The growth of the two pathogenic isolates was assessed using a medium containing eight fungicides; the results revealed varying degrees of inhibitory effects (Table 2). For H-4, trifloxystrobin·tebuconazole exhibited the strongest inhibitory impact, with an EC_50_ of 0.13 µg/mL, followed by difenoconazole·azoxystrobin (0.18 µg/mL) and difenoconazole (0.18 µg/mL). Azoxystrobin and lime sulfur exhibited the worst inhibitory effect, with EC_50_ values of 10.65 µg/mL and 5.29 µg/mL, respectively (Table 2). For H-5, the fungicidal effect was similar to that of H-4, with trifloxystrobin·tebuconazole being the most effective (EC_50_ of 0.14 μg/mL) and lime sulfur and azoxystrobin (3.59 μg/mL and 2.82 μg/mL, respectively) being the least effective (Table 2).

## 4. Discussion

Over the past two decades, the expansion of dragon fruit cultivation in Guizhou Province has led to a significant escalation in the severity of stem rot. *Fusarium proliferatum* was identified as the fungal pathogen of postharvest diseases on dragon fruits [6]. In this study, although a total of twenty-four fungal isolates were obtained from the infected dragon fruit samples, only two isolates of H-4 and H-5 were confirmed to be real pathogens based on Koch’s postulates. Using hyphae-based inoculation methods, we demonstrated that the remaining isolates were not pathogenic to dragon fruit, suggesting that they were most likely saprophytic or endophytic fungi on diseased tissues. It is also possible that some isolates are pathogens but some special conditions are required, such as complex infection.

Currently, the most effective method for confirming pathogenicity, following Koch’s postulates, involves the inoculation of live plants with a spore suspension [32]. Stem rot usually occurs under hot and humid weather conditions, as this elevated humidity provides an environment conducive to the formation, germination, and infection of plant tissues by fungal spores. Therefore, high humidity is considered an important factor in the occurrence of this disease [33]. Thus, we implemented a treatment involving sterile cotton impregnated with moisture to establish a suitable environment for disease occurrence.

A standard identification process combining molecular biology with traditional morphology was used for strain identification [34,35,36]. The results suggested that the pathogenic isolates H-4 and H-5 were the previously known pathogen, *F. oxysporum*, and a newly discovered pathogen, *F. concentricum*, respectively [37]. Both pathogens exhibited the ability to produce macroconidia and microconidia that can be disseminated through wind or rain, resulting in widespread disease outbreaks [38]. Furthermore, a comprehensive study of the pathogens was conducted using 95 different biochemical tests on FF microplates. This investigation revealed that although the types of carbon sources used by the two isolates differed, they demonstrated proficiency in utilizing a majority of the carbon sources, indicating that both isolates exhibited strong environmental survival abilities.

To effectively prevent and control this disease, we conducted laboratory experiments to evaluate the inhibitory effects of eight chemical agents on the growth of these pathogens. The results revealed that trifloxystrobin 25%·tebuconazole 50% (75 WG) exhibited the strongest inhibitory effect on both H-4 and H-5 isolates. This is supported by previous studies. Trifloxystrobin·tebuconazole was found to be one of the most effective fungicides with complete inhibition of mycelial growth of *Fusarium* species, including *F. oxysporum* and *F. concentricum* [39,40,41,42]. Difenoconazole 125 g/L·azoxystrobin 200 g/L (325 g/L SC) showed a slightly inferior inhibitory effect, but it was composed of two completely different active components compared with the former. Long-term, single-fungicide use is a significant factor contributing to pathogenic micro-organism resistance [43]. As a result, it is proposed that using these two fungicides interchangeably in field management will improve disease control. Notably, the concentration gradient of the experimental agents used in this study was only appropriate for laboratory screening; therefore, additional studies are required to determine the actual control efficacy and application concentration in field production. Furthermore, the experimental design for virulence determination focused on only a select few common chemical agents available in the market. Thus, additional research is warranted to explore their combined virulence determination, mixing proportions, and control effects [44]. Moreover, a comprehensive analysis of the impacts of the use of chemicals in field production on economic, social, and environmental benefits is imperative.

## 5. Conclusions

In this study, we investigated the causative pathogenic fungi responsible for dragon fruit stem rot in Guizhou Province. Following Koch’s postulates, we identified two isolates as the causative pathogens for this disease. Through morphological observations and molecular identification, we determined these two strains to be *F. oxysporum* and *F. concentricum*. Simultaneously, an experimental investigation using FF microplates showed that both isolates could utilize a wide range of carbon sources. To provide an important contribution to the field management and prevention of this disease, we examined the inhibitory effects of eight fungicides on both pathogens. Our findings revealed that 75% trifloxystrobin·tebuconazole exhibited the strongest inhibitory effect on the mycelial growth of the pathogens. These findings are promising in directing the efficient management of stem rot in dragon fruit in Guizhou, China.

## Figures and Tables

**Figure 1 jof-09-01178-f001:**
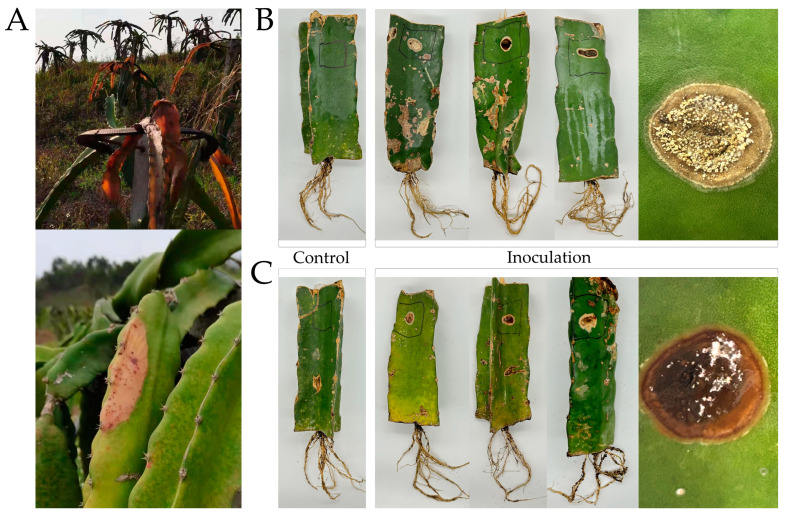
The symptoms of stem rot on dragon fruit. (**A**) Symptoms in field; (**B**) pathogenicity test results on stem inoculated with sterile water or the isolate H-4; (**C**) pathogenicity test results on stem inoculated with sterile water or the isolate H-5.

**Figure 2 jof-09-01178-f002:**
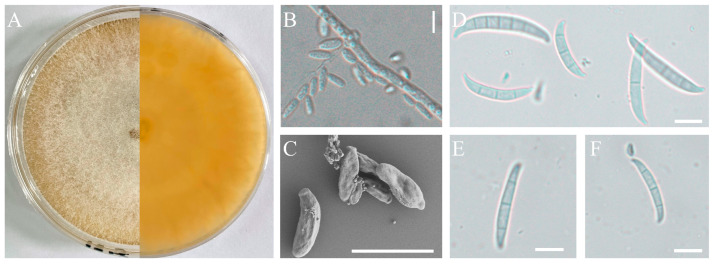
Morphological characteristics of the isolate H-4. Front and back of the colony (**A**); microconidia observed by optical microscopy (**B**) or scanning electron microscope (**C**); macroconidia observed by scanning electronic microscopy (SEM, SU8100, Japan) (**D**–**F**); scale bars: 10 µm.

**Figure 3 jof-09-01178-f003:**
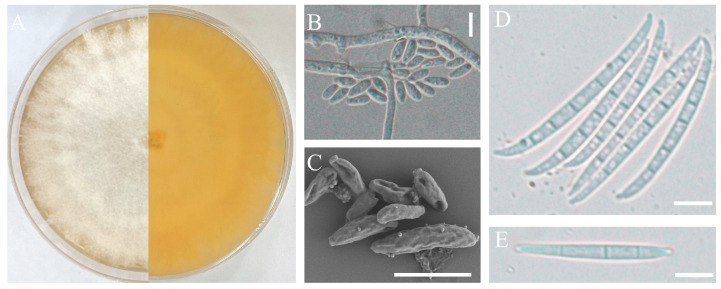
Morphological characteristics of the isolate H-5. Front and back of the colony (**A**); microconidia observed by optical microscopy (**B**) or scanning electron microscope (**C**); macroconidia observed by optical microscopy (**D**,**E**); scale bars: 10 µm.

**Figure 4 jof-09-01178-f004:**
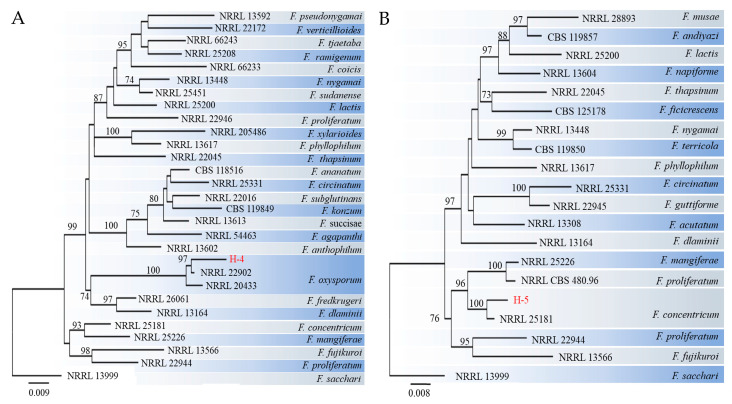
Maximum likelihood (ML) trees of the isolate H-4 (**A**) and H-5 (**B**) were generated based on internal transcribed spacer (ITS), translation elongation factor 1-alpha encoding gene (*EF-1α*), and the second largest subunit of RNA polymerase II encoding gene (*RPB2*) gene sequences. Numbers on the branches represent bootstrap values (BVs) greater than 70%. The red fonts showed the isolate H-4 or H-5.

**Table 1 jof-09-01178-t001:** Carbon source metabolism of the isoaltes on FF microplatasa.

Substrates	Isolates		Substrates	Isolates		Substrates	Isolates	
	H-4	H-5		H-4	H-5		H-4	H-5
Water	− *	−	a-Cyclodextrin	−	−	a-D-Glucose-1 -Phosphate	+	+
Tween 80	+−	+	ß-Cyclodextrin	−	−	N-Acetyl-ß-D-Glucosamine	−	+−
Glycerol	+−	+	Glucuronamide	+	−	D-Glucuronic Acid	+	+−
Dextrin	−	+−	i-Erythritol	+	+−	N-Acetyl-D-Galactosamine	+−	+−
Glycogen	+	+	D-Fructose	−	+−	N-Acetyl-ß-D-Man nosamine	−	−
Adonitol	+−	+−	L-Fucose	+	+−	m-Inositol	+−	+
Amygdalin	+−	+−	D-Galactose	−	+−	2-Keto-D-Gluconic Acid	+	+
D-Arabinose	−	+−	D-Galacturonic Acid	−	+−	a-D-Lactose	+	+−
L-Arabinose	−	+−	Gentiobiose	−	+−	Lactulose	−	−
D-Arabitol	−	+−	D-Gluconic Acid	+	+−	Maltitol	+−	−
Arbutin	+−	+	D-Glucosamine	−	+−	Maltose	−	+−
D-Cellobiose	−	+	a-D-Glucose	−	+−	Maltotriose	+−	+−
D-Mannitol	+	+	D-Ribose	+	+−	y-Aminobutyric Acid	+	+
D-Mannose	−	+−	Salicin	+	+−	Bromosuccinic Acid	+	+
D-Melezitose	+−	+−	Sedoheptulosan	+	+−	a-Methyl-D-Galactoside	+	−
D-Melibiose	+	+−	D-Sorbitol	+	+−	ß-Hydroxybutyric Acid	+	+
Fumaric Acid	+	+	L-Sorbose	+	+−	y- Hydroxybutyric Acid	+	+
L-Lactic Acid	+	+	Stachyose	+	+−	p-Hydroxy-phenylacetic Acid	+	+
D-Malic Acid	+	+	Sucrose	+	+−	a-Ketoglutaric Acid	+	+
L-Malic Acid	+	+	D-Tagatose	−	+−	D-Lactic Acid Methyl Ester	+	+
Quinic Acid	+	+	D-Trehalose	+	+−	ß-Methyl-D-Galactoside	+	+
D-Psicose	+−	+	Turanose	+	+−	a-Methyl-D-Glucoside	+	+−
D-Raffinose	+	+	Xylitol	+	+−	ß-Methyl-D-Glucoside	+−	+−
L-Rhamnose	+	+	D-Xylose	+	+−	Palatinose	−	+−
L-Proline	+	+	L-Alanine	+−	+−	L-Phenylalanine	+	+
Sebacic Acid	−	+	L-Alanyl-Glycine	+	+−	D-Saccharic Acid	+	+
Succinamic Acid	+	+	L-Asparagine	+	+−	L-Pyroglutamic Acid	+	+
Succinic Acid	+	+	L-Aspartic Acid	+	+−	Succinic Acid Mono-Methyl Ester	−	+
L-Serine	+	+	L-Glutamic Acid	+	+−	N-Acetyl-L-Glutamic Acid	+−	+−
L-Threonine	+	+	Gycyl-L-Glutamic Acid	+	+−	2-Aminoethanol	+	+
Uridine	+	+	L-Ornithine	+	+−	Putrescine	+	+
Adenosine	+	+	L-Alaninamide	+	+−	Adenosine-5′-Monophosphate	+	−

* “+” sensitive; “−” insensitive; “+−” weakly sensitive.

**Table 2 jof-09-01178-t002:** In vitro toxicity of eight fungicides against the isolates.

Fungicides	Concentrations of Active Ingredient (μg/mL)					EC_50_ (μg/mL)	
	C1	C2	C3	C4	C5	H-4	H-5
Azoxystrobin 250 g/L SC	1	2	4	6	8	10.65	2.82
Difenoconazole 125 g/L·azoxystrobin 200 g/L (325 g/L SC)	0.5	1	2	5	10	0.18	0.42
Difenoconazole 150 g/L·propiconazole 150 g/L (300 g/L SC)	0.2	0.5	1	2	4	0.38	1.06
Fludioxonil 25 g/L SC	1	2	4	6	8	1.15	0.96
Difenoconazole 40% SC	0.2	0.5	1	2	4	0.18	0.46
Trifloxystrobin 25%·tebuconazole 50% (75 WG)	0.2	0.5	1	2	4	0.13	0.14
Carbendazim 50% WP	0.01	0.05	0.1	0.5	1	0.89	1.10
Lime sulphur 45% WP	0.05	0.1	0.5	1	2	5.29	3.59

## Data Availability

Data are contained within the article.

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
