# Peer review of "Characterization and In Vitro Fungicide Sensitivity of Two Fusarium spp. Associated with Stem Rot of Dragon Fruit in Guizhou, China"

_jof, 2023, doi:10.3390/jof9121178_

Round 1
Reviewer 1 Report
Comments and Suggestions for Authors
The authors obtained 24 fungal pathogens from stem rot of dragon fruit. Based on morphological and phylogenetic analysis, two isolates were identified as F. oxysporum and F. concentricum. However, the identity of the other 22 isolates remains unknown. We don't know which species is predominant. Additionally, testing the sensitivities of only two isolates to various fungicides is not informative enough to understand the fungicide resistance situation in a given population of the pathogens in a specific area. Please read other reserchers' papers about the fungicide sensitivity study. Usually somewhere between 50 and 200 isolates, or even more isolates, are tested to certain fungicides.
While the scientific methods used in this study seem sound, the amount of data presented here is insufficient for publication. I would suggest that the authors should identify all 24 isolates and test their sensitivities to various fungicides to provide a more comprehensive understanding of the fungal pathogens.
Minor editorial suggestions and comments can be found in attached pdf file.

Reviewer 2 Report
Comments and Suggestions for Authors
The manuscript described the Fusarium pathogens of dragon fruit and analyzed their sensitivity to different fungicides. Two pathogens strains were identified as Fusarium oxysporum and F. concentricum respectively. In addition, the in vitro toxicity of eight fungicides against both pathogens were measured based on mycelium growth rate.
I have some comments on this manuscript as follow:
General comments
- Why were all the experiments carried out in the laboratory and the response was not confirmed at the field level?
- I think it would be better to adjust the title and mention that the experiments were carried out in vitro.
Abstract
- The introduction should go from the most general to the most specific, making it clear with the objectives of this investigation. The current structure of this section needs to be revised. Improving the abstract to provide the reader with the most relevant information in the manuscript is recommended.
Introduction
- The introduction is simple and superficial, as the authors should clarify whether there are previous works on the fungicides in plant Fusarium diseases control.
The authors should add a paragraph explaining the reason to use trifloxystrobin (FRAC 11), tebuconazole (FRAC 3), and difenoconazole (FRAC61 3) fungicides in controlling Fusarium diseases.
Result and discussion
- The authors must explain why they used the Biolog FF MicroPlate tests in their work. What specific characteristics did these tests show for Fusarium genus?
- The authors should provide photos of experimental results to clarify the differences between different fungicides treatments. Also, be careful with scientific names in italics.
- The discussion in this manuscript is very weak and needs to be improved.
Round 2
Reviewer 1 Report
Comments and Suggestions for Authors
The scientific methods employed in this manuscript appear sound. However, the sample size is inadequate for publication. If the stem rot is significant and caused by Fusarium spp., the authors should have examined more stem rot samples and obtained more isolates of the causal pathogen. Relying on just one isolate from each species may not provide a representative understanding of the entire situation. Furthermore, fungicide sensitivity varies from one isolate to another within the same species. The data from one single isolate is not informative at all.

Moderate editing of English language required
Author Response
Thank you for your valuable comments and suggestions.
Our text does indeed have the issue you raised, and we attach great importance to your feedback. But there are some issues that prevent us from making the best improvements. The explanation is as follows: the corresponding lack of a large number of strains. If we currently have a corresponding large number of strains, we would be more than happy to supplement the data because the workload is small, the time required is also short, and we can better gain your recognition. Unfortunately, due to our negligence, during the outbreak of the disease this year, we only collected a small number of samples, and only strains of H-4 and H-5 were left for further research, resulting in this deficiency.
But in order to make up for this issue as much as possible, we have had multiple discussions and ultimately decided to use relevant published papers as data support for the results of this article. We have searched a large number of literature and provided relevant descriptions in the introduction and discussion.
Best wish to you!
Reviewer 2 Report
Comments and Suggestions for Authors
The authors have significantly improved the text, the manuscript can be accepted in its present form.
Author Response
We appreciate your help and patience. Wishing you all the best!
Round 3
Reviewer 1 Report
Comments and Suggestions for Authors
The authors' answers to the reviewers' comments are satisfactory. However, the amount of work presented in this manuscript is more suitable for a shorter format such as "First Report", "Short Communication" or "Note" rather than a full research paper.
Please argue directly with the Senior Editors of the journal.
Comments on the Quality of English LanguageMinor editing of English language required.